# Rate-of-Kill (RoK) assays to triage large compound sets for Chagas disease drug discovery: Application to GSK Chagas Box

**Juan Cantizani, Pablo Gamallo, Ignacio Cotillo, Raquel Alvarez-Velilla[¤a], Julio Martin[¤b]***

Kinetoplastid DPU, Global Health R&D, GSK, Tres Cantos, Madrid, Spain

¤a Current address: Charles River Laboratories España, Sant Cugat del Valles, Barcelona, Spain
¤b Current address: Sciengement Lab Consulting, San Agustin del Guadalix, Madrid, Spain
* josejulio.martin@gmail.com

**Data Availability Statement:** All relevant data are within the manuscript and its Supporting information files.

## Abstract

Chagas disease (CD) is a human disease caused by *Trypanosoma cruzi*. Whilst endemic in Latin America, the disease is spread around the world due to migration flows, being estimated that 8 million people are infected worldwide and over 10,000 people die yearly of complications linked to CD. Current chemotherapeutics is restricted to only two drugs, i.e. benznidazole (BNZ) and nifurtimox (NIF), both being nitroaromatic compounds sharing mechanism of action and exerting suboptimal efficacy and serious adverse effects. Recent clinical trials conducted to reposition antifungal azoles have turned out disappointing due to poor efficacy outcomes despite their promising preclinical profile. This apparent lack of translation from bench models to the clinic raises the question of whether we are using the right *in vitro* tools for compound selection. We propose that speed of action and cidality, rather than potency, are properties that can differentiate those compounds with better prospect of success to show efficacy in animal models of CD. Here we investigate the use of *in vitro* assays looking at the kinetics of parasite kill as a valuable surrogate to tell apart slow- (i.e. azoles targeting CYP51) and fast-acting (i.e. nitroaromatic) compounds. Data analysis and experimental design have been optimised to make it amenable for high-throughput compound profiling. Automated data reduction of experimental kinetic points to tabulated curve descriptors in conjunction with PCA, k-means and hierarchical clustering provide drug discoverers with a roadmap to guide navigation from hit qualification of a screening campaign to compound optimisation programs and assessment of combo therapy potential. As an example, we have studied compounds belonging to the GSK Chagas Box stemmed from the HTS campaign run against the full GSK 1.8 million compounds collection [1].

## Author summary

One of the challenges in early drug discovery of small molecules is the improvement of the poor success rate in the translation from *in vitro* biological profile into efficacy in disease models, and ultimately in the clinic. Reductionist *in vitro* models on the bench may

**Funding:** Research published in this manuscript was supported by Wellcome Trust Award WT092340 (www.wellcome.ac.uk). The funders had no role in study design, data collection and analysis, decision to publish, or preparation of the manuscript. The authors received no specific funding for this work.

**Competing interests:** The authors have declared that no competing interests exist.

not properly recapitulate disease biology, thus overlooking critical properties of candidate compounds. Chagas Disease is a neglected tropical disease caused by *Trypanosoma cruzi*, a protozoan parasite with a complex life cycle. Despite the promising prospect based on *in vitro* and *in vivo* preclinical studies, efforts to reposition antifungal azoles turned out to be disappointing in clinical trials, with treatment failure in Chagas patients. This raises the question of whether we are using the right preclinical tools for decision-making about moving compounds forward for the treatment of this disease. We hypothesise that *in vitro* potency and efficacy values alone might be distorting the translational power of preclinical compounds, and we propose the use of rate-of-kill (RoK) assays in high-throughput mode. Herewith we disclose a simple, systematic, and automated methodology of analysis of the otherwise complex kinetic patterns, which provides drug discoverers with a navigation guide along a compound optimisation program or prioritisation of best exemplars across different chemical series.

## Introduction

American trypanosomiasis, also known as Chagas disease (CD), is an infectious human disease caused by *Trypanosoma cruzi* protozoan parasite. CD is endemic in Latin America where it is the major cause of death from cardiomyopathy. Additionally, if left untreated, it may cause serious gastrointestinal disorders or stroke. Although once geographically confined, the disease has since spread around the world because of migration flows. It is estimated that 8 million people are infected worldwide and over 10,000 people die yearly of complications linked to CD [2]. Moreover, 75 million people live at risk of contracting CD just in Latin America, and it is estimated that only 10% of the actual infected population is diagnosed [3]. The disease is usually transmitted by insects of the subfamily *Triatominae*, so-called ´kissing bugs´. Noteworthily, vertical congenital transmission remains a significant way CD is transmitted. Human CD progresses through three phases, i.e. acute, indeterminate and chronic symptomatic [4]. The acute phase may last up to two months, with minor or no symptoms. Then, the immune response of the individuals can control infection but do not completely clear parasite nor prevent eventual development of the disease. Patients may remain asymptomatic or indeterminate for 10–30 years, and just about 30–40% of them will ultimately evolve to chronic phase with cardiac or digestive clinical manifestations.

Current chemotherapeutic armamentarium is restricted to only two drugs, i.e. benznidazole (BNZ) and nifurtimox (NIF), both being nitro-aromatic compounds which exert their anti-trypanosomal effect upon intracellular activation by specific parasite nitroreductases [5]. Treatment is indicated for patients in the acute or early chronic phases and infants with congenital infection. Efficacy of treatment decreases as the duration of the infection lengthens, and the risk of adverse effects increases with age [2]. Adults with the indeterminate form of the disease should be treated, but its potential benefits in preventing or delaying the development of CD should be weighed against the long duration and frequent adverse events. BNZ is contraindicated in the case of pregnancy and kidney or liver failure, whereas NIF is not recommended in patients with psychiatric or neurological disorders. Adverse effects of BNZ are one the main disadvantages of using this drug, the most common one observed being dermatological alterations followed by gastrointestinal disturbances [6]. These effects lead to interruption of treatment in up to 20–30% of patients after an average time to interruption of 26.7 days [7].

All in all, new low-cost drugs that are safer, more efficacious and with shorter duration of treatment are needed. In this regard, and despite the promising prospect based on *in vitro* and

*in vivo* preclinical studies, efforts to reposition antifungal azoles (i.e. Posaconazole and the Ravuconazole prodrug E1224) turned out to be disappointing in clinical trials, with treatment failure in Chagas patients reaching 70% to 90%, as opposed to 6% to 30% failure for BNZ-treated patients [4, 8–10]. Some authors have argued an inadequate dosing of Posaconazole (i.e. 10–20% of curative dose in mice of 20 mg/Kg/d) as the reason for failure [8, 11–14]. The lack of translation from bench models to the clinic observed for the azoles raises the question of whether we are using the right preclinical tools and decision-making processes to move compounds forward for the treatment of this disease. For instance, are these models properly recapitulating the PK/PD in clinical setting, so that we are looking at the relevant features of candidate compounds? It seems obvious that potency and efficacy values alone might be distorting the translational power of preclinical compounds.

*in vitro* assays looking at the kinetics of parasite kill, in conjunction with *in vivo* studies, can help to identify whether a compound is fast- or slow-acting and figure out the PK/PD properties of a compound, which may ultimately guide projections on the speed of parasite killing in humans to get a first indication of the treatment duration needed.

Rate-of-kill (RoK) kind of assays have been well defined for other parasitic diseases such as human African trypanosomiasis (i.e. *Trypanosoma brucei*) [15], leishmaniasis (*Leishmania spp.*) [16] and malaria (i.e. *Plasmodium spp*) [17], but not so extensively described for CD (i.e. *Trypanosoma cruzi*). Compounds belonging to different chemical classes and exerting their anti-trypanosomal action through different mechanisms and molecular targets can also exhibit a differential behaviour in *in vitro* RoK assays against a panel of *T. cruzi* strains. Although less potent, the nitroaromatics and the oxaboroles showed broad efficacy against all *T. cruzi* tested and were rapidly trypanocidal, whilst ergosterol biosynthesis inhibitors, such as azoles inhibiting parasite CYP51, exhibited variable activity that was both compound- and strain-specific. Furthermore, they were unable to eradicate intracellular infection even after 7 days of continuous compound exposure at most efficacious concentrations [18]. Despite their high potency, all CYP51 inhibitors exhibited a moderately slow killing profile and a relatively high number of infected cells remained at the end of a static-cidal assay time-course, unlike nitro-aromatic compounds whose mode of action result in fast, replication-independent parasite death [19]. From the drug discovery perspective, a fast-acting and long-lasting compound would be optimal. Benznidazole is certainly a fast-acting compound, and it is questionable whether a slow-acting compound is desirable.

Herewith we present the application of RoK assays to characterise the potential anti-chagasic compounds with antitrypanomal activity against *T. cruzi*. These assays when run with intracellular amastigotes can tell apart chemical classes exhibiting different mode of actions, such as azoles and nitroaromatics. Data analysis and experimental design have been optimised to make it amenable for high-throughput compound profiling. We show its use for the differentiation and prioritisation of compounds within the GSK Chagas Box (aka TCKAS) [1]. We propose that speed of action and cidality, rather than potency, are properties that can differentiate those compounds with better prospect of success to show efficacy in animal models of Chagas disease.

## Methods

### Parasites and mammalian cells

LLC-MK2 (green monkey kidney epithelial cells) and H9c2 (rat cardiomyocytes) cell lines were cultured in DMEM (Life-Technologies) supplemented with 10% FBS (Biowest, USA), 100 U/ml penicillin (Sigma-Aldrich), 100 µg/ml streptomycin (Sigma-Aldrich), and 4 mM or 2 mM L-glutamine (Sigma-Aldrich), respectively. Both cell lines were purchased at the

European Collection of Cell Cultures (ECACC, Salisbury, UK) and were grown in 225 cm$^3$ T-Flasks (Corning) at 37˚C, 5% $CO_2$ and >95% humidity. The DMEM formulation for the assay lacked phenol red (Life-Technologies reference 31053) and was supplemented with 2% FBS, 100 U/ml penicillin, 100 μg/ml streptomycin, 2 mM L-Glutamine, 1 mM sodium pyruvate (Life-Technologies), and 25 mM HEPES (Life-Technologies) [20]. *T. cruzi* Tulahuen strain was kindly provided by Dr. Buckner (University of Washington, Seattle, USA) [21] and maintained in culture by weekly infection of LLC-MK2 cells in the same DMEM formulation used for cell growth, but supplemented with 2% FBS. Trypomastigote forms were obtained from the supernatants of LLC-MK2 infected cultures harvested between days 5 and 8 of infection.

## Compound preparation

Assay plates were pre-dispensed with the compounds dissolved in neat DMSO by using an Echo liquid handler (Labcyte Inc.). The primary screening was performed at single shot at a final concentration *per* well specified for each assay. Further on, to determine compounds´ potency 11-concentration points in a 1:3 dilution pattern was stamped starting at 10 times the concentration used for the single shot assay. DMSO at the final same concentration of the compounds was used as Control 1 for each assay. Plates were stored frozen at -20˚C until used, when they were allowed to equilibrate at room temperature before adding the assay mixture in.

## *T. cruzi* trypomastigote assays

*T. cruzi* Tulahuen trypomastigotes obtained from the supernatants of LLC-MK2 infected cultures in a 225 cm$^2$ T-Flasks harvested between days 5 and 8 of infection. Trypomastigotes ($5 \times 10^4$ per well) in 50 μl of DMEM/ 2% FBS were dispensed into white 384 plates (Greiner) pre-stamped with 250 nl compound a single-shot final concentration of 5 μM, or in dose response. Plates were incubated at 37˚C in 5% $CO_2$ for 48 or 72 hours. Cell Titer-Glo cell viability solution (Promega) was added to each well (50 μL) incubated at room temperature for 10 min and luminescence read by Envision plate reader (Perkin Elmer). This assay quantifies the amount of ATP present which is directly proportional to the number of metabolically active cells. All data were normalised (100% effect control for no cells and 0% effect control for DMSO). This assay was adapted from MacLean et al., 2018 [22]. Fig 1 depicts the results obtained for this assay.

## Rate-of-Kill (RoK) assay

This assay was performed both at single shot with a final concentration of 5 μM and at dose-response curves of 11 points with 1/3 serial dilution started with an initial concentration of 50 μM.

For this high-content intracellular imaging assay, H9c2 cells were seeded in 225 cm$^2$ T-FLASK in DMEM with 10% FBS for 4 h to allow attachment. *T. cruzi* trypomastigotes, collected at days 5 to 8 after infection from LLC-MK2 parasite infected cultures, were allowed to swim out for 4 h at 37˚C from a centrifuged pellet (1,600 x g / 10 min). Trypomastigotes were then collected and counted in a CASY Cell Counter using the 60 μm capillary. Trypomastigotes, in supplemented DMEM, were added to H9c2 cultures in a multiplicity of infection (MOI) of 8 and incubated for 18 h. Cells were washed once with PBS before incubation of the infected H9c2 monolayer with trypsin (Life-Technologies) to detach cells from the flask. Cells were counted in a CASY Cell Counter, using the 150 μm capillary, and their density set at 5 x $10^4$ cells/mL in supplemented DMEM. Infected H9c2 were dispensed into 384-well plates at

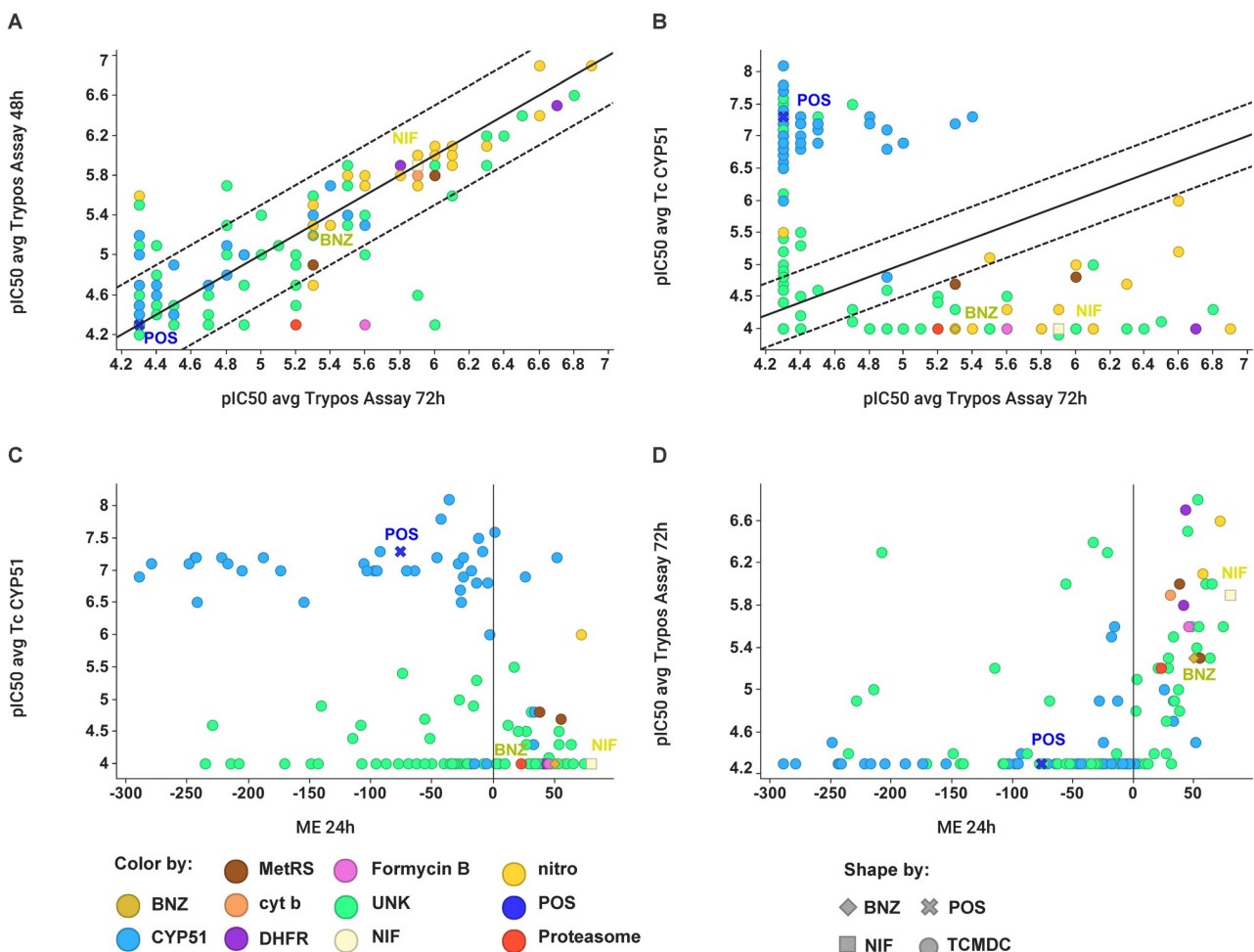

**Fig 1. Correlation analysis of activity across amastigotes, trypomastigotes and TcCYP51 enzyme for compounds in GSK Chagas Box.** (A) pIC50 average values in trypomastigotes assay at two different incubation times, i.e. 48 hours (Y axis) and 72 hours (X axis). (B) pIC50 average for TcCYP51 inhibition *versus* activity against trypomastigotes at 72 hours. (C) pIC50 average of TcCYP51 inhibition *versus* activity against intracellular amastigotes as maximum effect at 24 hours (ME 24h). (D) pIC50 values in trypomastigotes assay at 72 hours *versus* activity against intracellular amastigotes as maximum effect at 24 hours (ME 24h). For all scatter plots colours correspond to a putative target identification based on chemical structures. For comparison purposes NIF, POS and BNZ were included and marked.

50 µL per well using a Multidrop Combi dispenser. Control wells used to determine a 100% parasitic growth (full 6th column of each plate) were left untreated, whereas not infected cells were added in the control wells used as 0% parasite growth (or 100% parasite growth inhibition; full 18th column of each plate). The four plates corresponding to each RoK assay were predispensed for the same mother plate. After seeding them, each plate was incubated at 37˚C, 5% $CO_2$ for 4, 24, 48 or 72 hours. Cultures were then fixed and stained with 50 µL per well of a PBS solution containing 4% formaldehyde and 2 µM DRAQ5 DNA dye (BioStatus, UK). Then plates were read in a PerkinElmer Opera Phenix using a 20x objective, 3 fields per well. DRAQ5 signal was detected using 640 nm excitation laser and a 690/50 nm emission detection filter. Automated image analysis was performed with a script developed on Acapella High Content Imaging and Analysis Software (PerkinElmer). Two outputs were provided for each sample well: (1) number of host cells nuclei to determine drug-related cytotoxicity (aka H9c2) and (2) number of amastigotes per cell as infection level measurement (aka AM/CELL). Each

output is plotted as a time course for each compound at every concentration. This assay was adapted from Alonso-Padilla *et al.* [23] and Peña *et al.* [1].

## RoK combination assay

The assay was performed following the same experimental procedure mentioned above in the RoK assay. The infected H9c2 cells were treated with final 10 and 50 μM concentrations of BNZ and DMSO before being dispensed into the plates containing the compounds. For the combination with 50 μM BNZ, two plates were incubated for 4 and 24 hours respectively before being fixed. For the combinations with 10 μM BNZ and with DMSO, four plates for each condition were incubated for 4, 24, 48 and 72 hours before being fixed.

## Curve descriptors

To identify and classify the different compounds, the number of amastigotes *per* cell (AM/ CELL) raw data obtained in rate of kill assay are normalised using the average of C1 (infected cells) at time 4 hours as 0% effect and the average of C2 (non-infected cells) for each time as 100% effect, according to the following Eq 1:

$$\% \, CIDAL_{RESPONSE} = 100*[Avg\_C1(4h) - RawData]/[Avg\_C1\,(4h) - Avg\_C2] \tag{1}$$

As result of this normalisation, we have defined a series of curve descriptors, which qualify the RoK pattern based on effect (parasite kill), potency (compound concentration) and speed of action (time):

- **ME (Maximum Effect)** is the maximum percentage of parasite burden reduction reached by any compound concentration. Compounds that decrease the initial parasite load will show a positive value of ME, whereas compounds that allow the progression of infection will render a negative ME value. Likewise, compounds that maintain the initial parasite burden will have a ME value equals to zero. ME have been calculated for 24, 48 and 72 hours.

- **MCC50 (Minimal Cidal Concentration 50%)** is the lowest compound concentration that reduces 50% of the parasite load measured at zero time.

- **MCCE (Minimal Compound Concentration with Effect)**. By using the Area-Under-the-Curve (AUC) of the number of amastigotes *per* cell *versus* time as metric, this is the lowest compound concentration that decreases the AUC by at least 50% with respect to the C1 control (i.e. infected cells).

- **t50%** is the time elapsed at one particular concentration to reduce 50% of initial parasite load. It is calculated for the MCC50 (**t50%_MCC50**) and for the maximum concentration (**t50%_Cmax**). Only compounds able to reach a percentage of inhibition equal or greater than 50% render a data and no data are obtained otherwise (ND: not determined). The value of this parameter is not either interpolated or extrapolated, and it refers to the first experimental time-course point at which a 50% of effect is reached or exceeded.

- **tlag** defines the time elapsed for one particular concentration to reduce parasite burden between two consecutive points at any moment of the time course. It is calculated for the MCC50 (**tlag_MCC50**) and for the maximum concentration (**tlag_Cmax**). The value of this parameter is not either interpolated or extrapolated, and it refers to the first experimental time-course point of a positive slope in the normalised plot (i.e. %Cidal_Response).

- **Cmin_Tox (Minimal CytoToxic Concentration)** is the lowest compound concentration that is reducing the number of host mammalian cells below 50% of initial cell density.

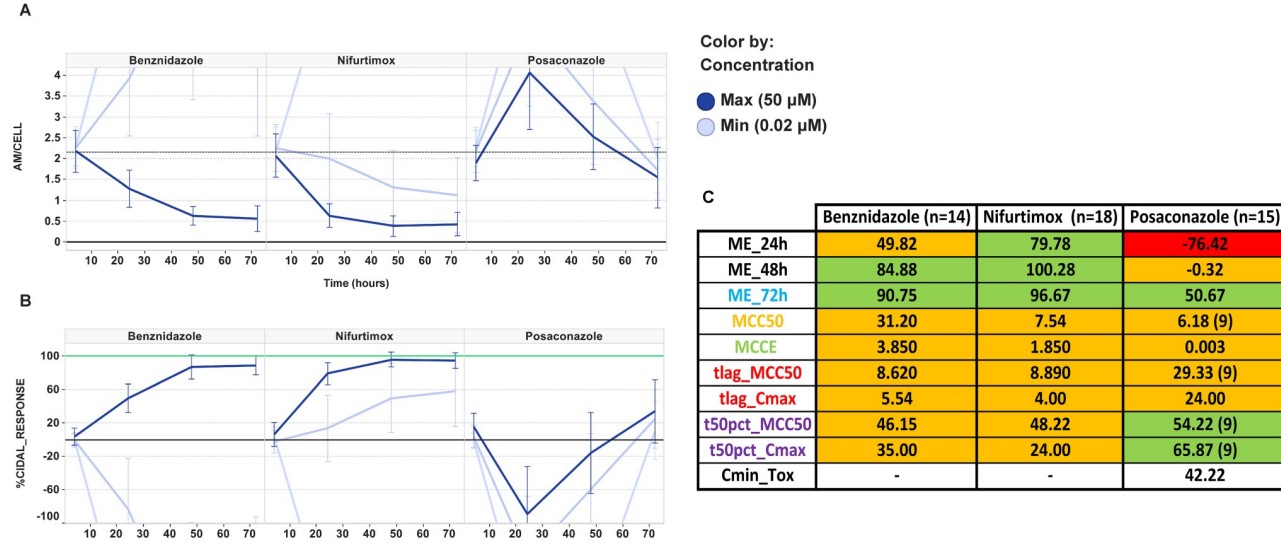

**Fig 2. Experimental Rate-of-Kill time courses and curve descriptors for Benznidazole (BNZ), Nifurtimox (NIF) and Posaconazole (POS).** (A) Average number of *T. cruzi* amastigotes *per* host cell (AM/CELL) over time in hours. Errorbars represent the standard deviation of the average for all the independent experiments indicated in the table of panel C. (B) AM/CELL data are normalised and transformed to "%CIDAL_RESPONSE", as defined in Methods. Error bars represent the standard deviation of the average for all the independent experiments indicated in the table of panel C. (C) RAG table of curve descriptors for BNZ, BNZ and POS. Thresholds for colours are set as follows: **ME**, green for values higher than 50, yellow for values between 50 and -10 and red for values lower than -10; **MCCE**, green colour for values lower than 2 µM, yellow for values between 2 and 6 µM and red for values higher than 6 µM; **MCC50**, green for values lower than 5, yellow for values between 5 and 16 and red for values higher than 16; **t50% _MCC50, t50%_Cmax** and **tlag_Cmax**, green colour were used for values lower than 24 h, yellow for values between 24 and 48h and red for values higher than 48h; **tlag_MCC50** green for values lower than 24 h, yellow for 24h and red for values higher than 24h. Values in brackets at the column headers indicate the number of experiments included in the analysis; and values in brackets inside the cells correspond to the number of experiments that meet the acceptance criteria of activity within time and concentration ranges used for computational calculations of each parameter (note: no figure in brackets mean that all experiments indicated in column header were computed).

---

Cytotoxic compounds may mislead a *bona fide* antiparasitic action. Therefore, this descriptor is used to flag this kind of compounds and warn about the genuine meaning of the rest of descriptors.

Visual descriptions and cases for Benznidazole, Nifurtimox (as two exemplars of the nitroaromatic chemical class) and Posaconazole (as exemplar of the azole chemical class) are depicted in Fig 2.

The computation of these descriptors was performed by means of a script implemented in the programming language R [24]. The program coding in R language is included as Supporting Information S1 Code. The descriptors for each compound were computed using the concentration range below its Cmin_Tox.

## Principal component analysis (PCA)

PCA uses the covariance (or correlation) matrix of the variables to generate new variables, which are linear combinations of the originals but that are uncorrelated between them (the so-called "principal components") [25]. The principal components are ordered by the variation explained and, ideally, the first two or three principal components account for most of the variation of the original data.

The purpose of applying PCA to these RoK descriptors is exploratory and aims at assessing its ability to discriminate and classify compounds with similar activity. The PCA analysis and

its visualizations were performed, respectively, with the packages 'FactoMineR' [26] and 'factoextra' [27], available in the programming language R.

Since PCA does not deal with missing values for the variables, for those compounds whose descriptors tlags, t50s, MCC50 and/or MCCE are not rendered within the tested timepoints and concentrations, they are set respectively to the maximum tested, that is, undefined tlags to 72h and undefined MCC50 and/or MCCE to the maximum concentration tested for the corresponding compound. To make all the variables comparable and avoid the effects of different scales, all the variables are standardised (mean zero and standard deviation one) prior to performing the PCA.

### k-means clustering

k-means clustering was performed using the module available in Spotfire (http://spotfire.tibco.com/). Using the calculated **Cmin_Tox** value, the highest concentration without effect in the host cells have been selected for each compound for the clustering. The Euclidean distance between the RoK curves was used as similarity measure and six clusters have been defined [28].

### Hierarchical clustering

By considering the list of tested compounds (as rows) and their corresponding curve descriptors (as columns) arranged as a matrix (with its corresponding heat map visualisation), the hierarchical clustering tool implemented in TIBCO Spotfire [29] was applied and the results represented by means of a dendrogram (tree-structured graph). The clustering method considered was 'Complete Linkage', the Euclidean metric was used as similarity measure and the curve descriptors were normalised by mean prior to clustering.

### Biosafety

Experimental work with live *T. cruzi* cells was carried out following standard operating procedures in compliance with biosafety level 3 regulations (BSL3).

Data of the boxes are available at http://ebi.ac.uk/chemblntd, CYP51 data used for this manuscript are included in the original boxes paper [1].

## Results

### Trypomastigotes assays with Chagas Box compounds

The compounds of the GSK *Chagas* Box from TCAKS (i.e. Tres Cantos Anti-Kinetoplastids Set) [1] were assayed in trypomastigotes assay at two different incubation times, i.e. 48 and 72 hours. Z-prime values of the assay were higher than 0.6 for all plates and POS, BNZ and NIF were included as standards in all the assays. Results are shown in S1 Table. Most of the active compounds in the trypomastigotes assay correlates well at both times in terms of potency without discrepancies (Fig 1). 136 out of 222 compounds have a pIC50 lower than 5 (10 μM) at the two incubation times. In general terms, compounds with activity against CYP51 assay above pIC50 of 5 have no activity in the trypomastigotes assay (Fig 1B).

### Rate-of-Kill (RoK) assay

The results of POS, BNZ and NIF are collected in Fig 2 and S2 Table. POS exhibits a differential kinetic pattern from nitroaromatic compounds, such as NIF and BNZ (Fig 2), which results in distinct sets of curve descriptors values (S2 Table). Main differences are: (a) lower ME, i.e. POS is not able to clean up the parasites from the culture, (b) longer tlag_MCC50 and

tlag_Cmax values, i.e. POS requires much more time to exert a parasiticidal effect and reduce parasite load, and (c) BNZ and NIF are less potent than POS and show higher values of MCC50 and MCCE. Therefore, these descriptors enable the differentiation between these two chemical classes. This result correlates with previous *in vitro* [22] and *in vivo* studies [30, 31] and ultimately with the outcome of the clinical trials with azoles and nitro aromatic compounds [8]. The differentiation of BNZ and NIF from POS throughout RoK and trypomastigotes assays leads us to propose these assays as predictors of efficacy in animal models, and eventually in the clinic.

110 compounds out of Chagas Box were tested in the RoK assay. The selection of the compounds has been made after a chemical diversity assessment. The standards were included in each experiment and have been used as control for QC of the pharmacology and reproducibility of the assay. POS, BNZ and NIF behaved in each experiment as aforementioned. S2 Table contains the descriptors for all the compounds.

ME is a key descriptor in the RoK curve because it may give an indication of the ability of the compound to fully clean up parasites from culture, thus avoiding relapse upon compound wash-out [19]. ME calculated at 24 hours captures the compounds with fast speed of cidal action. This descriptor by itself may suffice to tell apart azoles and nitro-aromatic compounds. ME at longer times, such as 48 or 72 hours, does not allow to make this differentiation since CYP51 inhibitors exhibit bell-shape kinetics and cidal effect is incipient then. Noteworthy, cytotoxicity along time may affect and distort the estimation of curve descriptors and interpretation.

The ME descriptor considers the effect on the different parasites forms present, including persisters or quiescent forms. Compounds that do not achieve ME close to 100% will likely not be efficacious enough to avoid relapse or achieve complete cure in more complex and longer assays where the effect of these parasitic forms have more relevance [22]. In general terms, compounds acting by CYP51 inhibition, such as azoles, render negative ME values at 24 hours, which indicates static, rather than cidal, effect at short incubation times (Fig 1C). Moreover, positive ME values at 24 hours, corresponding with fast-acting killing compounds, correlate with activity against trypomastigote forms (Fig 1D). This observation points to the value of trypomastigote assay, which measures the anti-trypanosomatidal effect on a non-replicating and metabolically low parasite form, as a surrogate assay to flag compounds with cidal mechanism of actions.

Two potency descriptors of the curves have been defined, i.e. MCC50 and MCCE, which complement each other. For instance, some CYP51 inhibitors are not able to reduce the initial parasite burden more than the 50%, so that MCC50 cannot be calculated and they are annotated as ND (i.e. not determined) in S2 Table. However, this compound class usually exhibits low MCCE values because its high potency, so that after 72 hours several concentrations can reduce the initial parasite load (Fig 2).

As for speed of action, we propose the tlag parameters, which are related to the time needed for the compound to start causing a cidal effect and have positive slope (Fig 2B and 2C). As aforementioned, high tlag_Cmax values clearly differentiate POS and CYP51 inhibitors from other compound classes. Likewise, t50%_Cmax differs quite significantly between azoles and nitroaromatic drugs, with shorter times being estimated for nitroaromatic drugs in comparison with azoles and CYP51 active compounds.

Although azoles are usually potent compounds against *T. cruzi*, these compounds are more slowly acting than nitroaromatic drugs. Some studies suggest that several replication cycles of the parasite are required to see an effect in the presence of CYP51 inhibition [22, 32]. Therefore, we propose that a comprehensive analysis and comparison of RoK behaviour allow us to

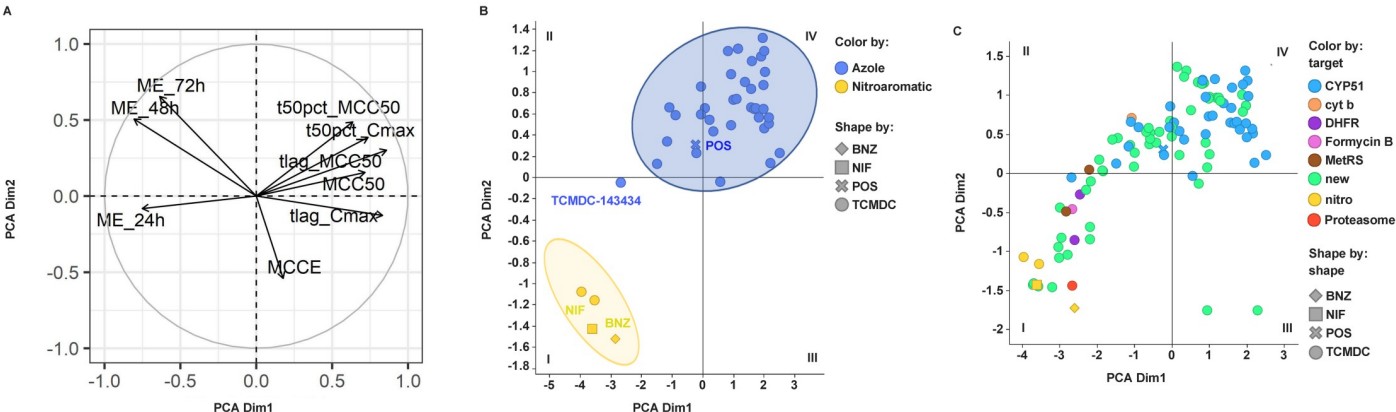

**Fig 3. Clustering by principal component analysis (PCA) of RoK curve descriptors.** (A) Variable correlation plot for the first two principal components. (B) 2D PCA mapping for azoles in blue and nitroaromatic compounds in yellow. The ellipses represent an estimated 95% confidence area for each group representation. (C) PCA analysis for GSK Chagas Box compounds coloured by putative mode of action. UNK, unknown mode of action. Points shape has been used to mark reference compounds (i.e. BNZ, NIF and POS) and compounds that belong to the GSK Chagas box.

get a more informative and better predicting indication of the prospect of the compounds to translate *in vitro* activity into *in vivo* models of efficacy.

## Compound clustering analysis

**PCA (Principal component analysis) clustering.** One method to reduce the dimensionality of this multivariate data, when the variables are quantitative and interrelated, is to apply Principal Component Analysis. The representation of the variables (RoK descriptors) on the plane of the first two principal components is shown in Fig 3A, together with the correlation circle. The two first principal components of the RoK descriptors account for almost a 70% of the variation. The proximity between ME_48h and ME_72h (maximum effects at 48 and 72 hours) indicates positive correlation between them. Closeness between tlag_MCC50 and both t50%s indicates positive correlation between them. The Scree plot and the contribution of the variables to first two principal components are shown in the Supporting Information (S1 Fig with Scree plot and S2 Fig with variables contribution). Dimension 1 mainly recapitulates features related with speed of action, e.g. tlag descriptors, whereas dimension 2 predominantly gathers features linked to effect or potency, e.g. ME and MCCE.

The representation of the compounds in plane defined by the two principal components is shown in Fig 3B. By using a different colour to represent the compounds belonging to either the azoles (i.e. CYP51 inhibitors) group (in blue) or the nitroaromatic group (in yellow), the plot demonstrates that PCA is able to differentiate appropriately both groups. The overlaid elliptical regions are the 95% confidence regions computed from the coordinates of the compounds of each group (assuming a bivariate normal distribution) [27]. Noteworthy, only one compound, TCMDC-143434, within the azoles class maps outside their corresponding confidence area. However, TCMDC-143434 is an atypical CYP51 inhibitor showing hints of cytotoxicity even at low concentrations, suggesting a more promiscuous mode of action. Fig 3C displays the scatter plot of all compounds included in the current study on the 2D PCA plot.

**k-means clustering.** Fig 4A shows the output of a k-means clustering of the full RoK experimental curves. Highest non-cytotoxic concentration of each compound is considered in the analysis. Six clusters turned out to be a reasonable number to differentiate and group akin compounds within the diverse range of kinetic behaviours. On one hand, most of the compounds (82%) belonging to the CYP51/azole class are scattered throughout clusters 1, 3, 4 and

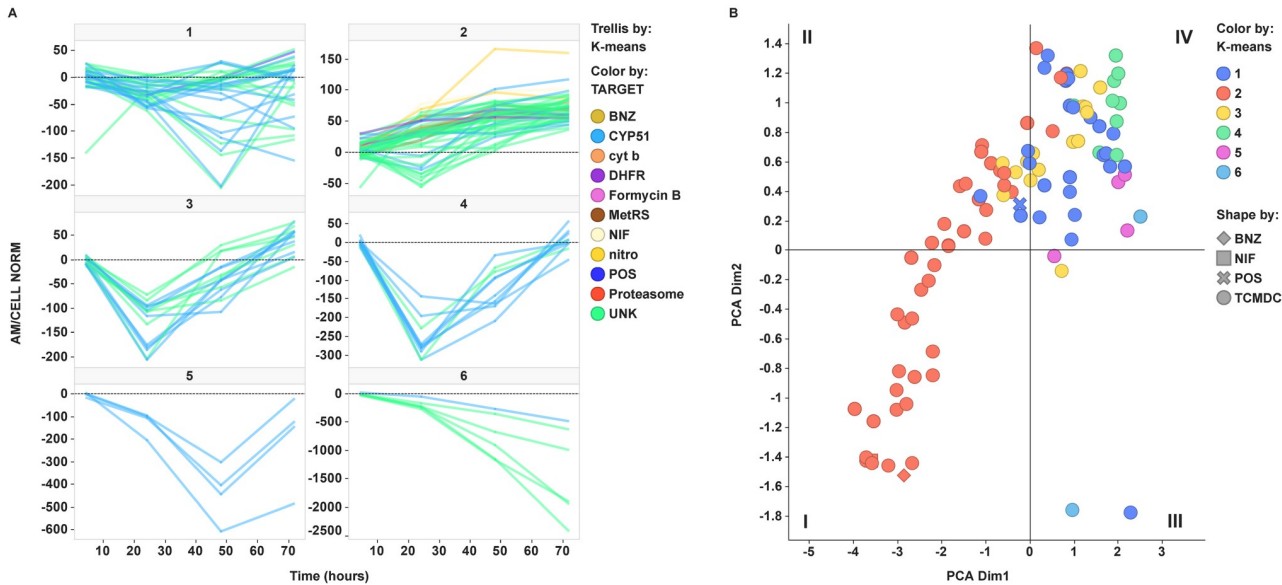

**Fig 4. k-means clustering for RoK curves.** (A) Trellis plot by cluster. Non-cytotoxic concentration was selected for each compound based on Cmin_Tox value. Compounds are coloured by the putative mode of action of the compounds. UNK, unknown mode of action. Reference compounds for nitro (BNZ and NIF) and azole (POS) compound classes have been included in the colour code in yellow and blue respectively. (B) Relationship between PCA and k-means clustering. Dots representing compounds are coloured by k-means cluster number. Points shape has been used to mark reference compounds and compounds that belong to the GSK Chagas box.

5. All these clusters share a pattern with a minimum compatible with an initial static behaviour which eventually becomes cidal at longer times, but they can be told apart according to the situation of the minimum and the slopes of the ascending and descending segments of the time-course. On the other hand, nitroaromatic compounds exemplified by nifurtimox and benznidazole falls in cluster 2, which is mostly populated by curves with a noticeable cidal action from zero time. Noteworthy, a minority of putative CYP51/azole class (18%) also fall into cluster 2. Inactive or barely active compounds are grouped in cluster 6.

Both PCA and k-means clustering are correlated. Relationship between PCA and k-means clustering outputs is depicted in Fig 4B.

**Hierarchical clustering.** RoK descriptors were also subjected to hierarchical clustering and the outcome depicted as a tree-structured dendrogram connected to a heat map visualisation (Fig 5). Row dendrogram shows the Euclidean distance between compounds and which nodes each compound belongs to because of clustering. Interestingly, hierarchical clustering of compounds is in good agreement with PCA and k-means clustering. Noteworthy, compounds from quadrants IV (where CYP51 inhibitors predominantly fall) and I (where most of nitroaromatic compounds are located) in Figs 3C and 4B are clustered at highest Euclidean distance, which nicely validates how hierarchical clustering might guide compound selection and optimisation.

## Single shot RoK screening assay

A comparison of RoK and trypomastigotes assays have been made using a set of 4,000 compounds of GSK collection. The compounds have been assayed at one single concentration and one single incubation time, i.e. 5 µM and 24 hours incubation. As aforementioned, this time-point tells apart CYP51 inhibitors and cidal compounds.

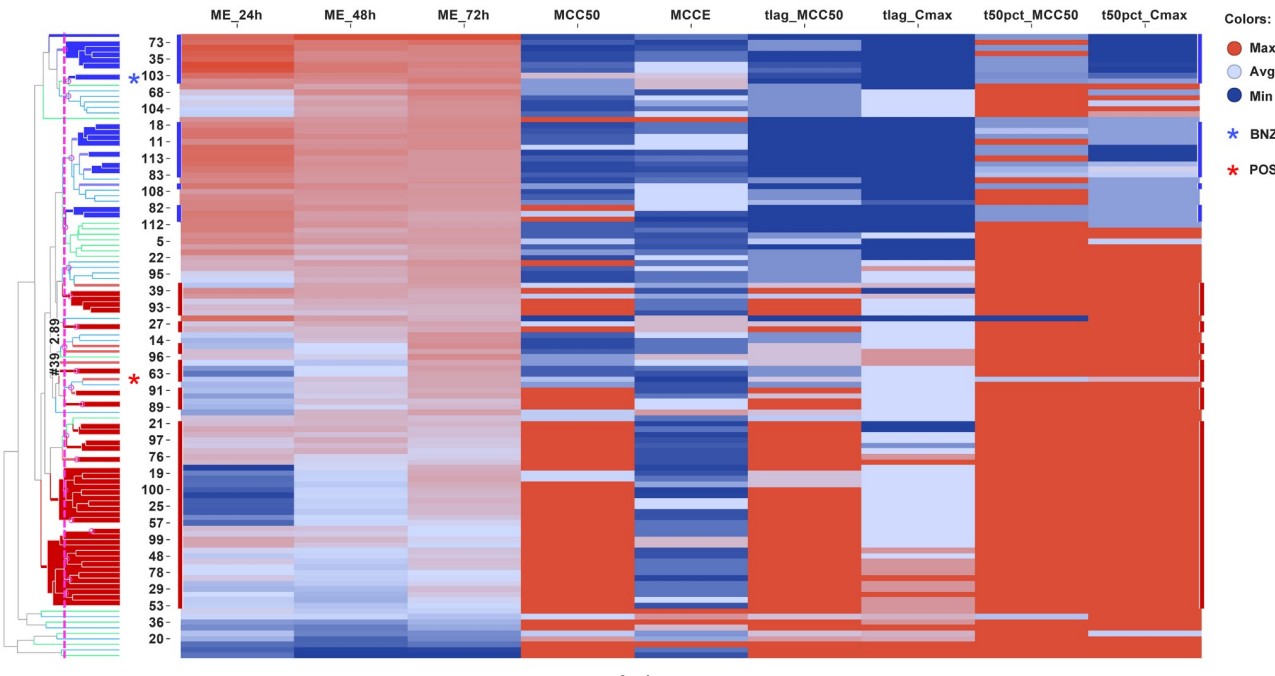

**Fig 5. Hierarchical clustering for RoK curves of Chagas Box compounds.** Dendrogram for compounds in rows is depicted connected to heat map based on the nine curve descriptors in columns. Compounds in the row dendrogram are coloured as follows: in blue, compounds falling in quadrant I of Fig 3C; in red, compounds falling in quadrant IV of Fig 3C. Dotted pink line corresponds to the pruning line indicating number of clusters and Euclidean distance.

By correlating activity against intracellular amastigotes and cytotoxicity *versus* host cells, three groups of compounds were differentiated (Fig 6A): in red the inactive compounds, in blue the putative CYP51 inhibitors based on the bell-shape described for these compounds in the RoK curves, and in green the cidal compounds with no cytotoxicity effect at 24 hours.

As previously described, there is a high number of CYP51 inhibitors predominantly populating the compounds set, which can be easily distinguished and flagged out from the rest of compounds. The chemical structure of the compounds has been reviewed and contains several motifs described as CYP51 inhibitors.

The compounds highlighted in green in the Fig 6A and 6B, have been selected based on the activity higher than zero in the normalised AM/CELL value and in the non-toxic effect against the host cells with a total number of cells higher than 150. 87.5% of compounds tested in RoK showed ME_24h values higher than 50%, as an indicator of fast cidality, validating the experimental approach for single-shot screening.

Since CYP51 inhibitors are slow-acting *versus* intracellular amastigotes and trypomastigote-inactive, a scatter plot was generated for activities on intracellular amastigotes and trypomastigotes (Fig 6B). As previously mentioned, correlation between both activities is hardly apparent.

## Combinations studies

BNZ has been selected as a partner for combinations studies. BNZ was added to the cells prior to dispensing into the plate containing the serial dilution of the compounds and then RoK

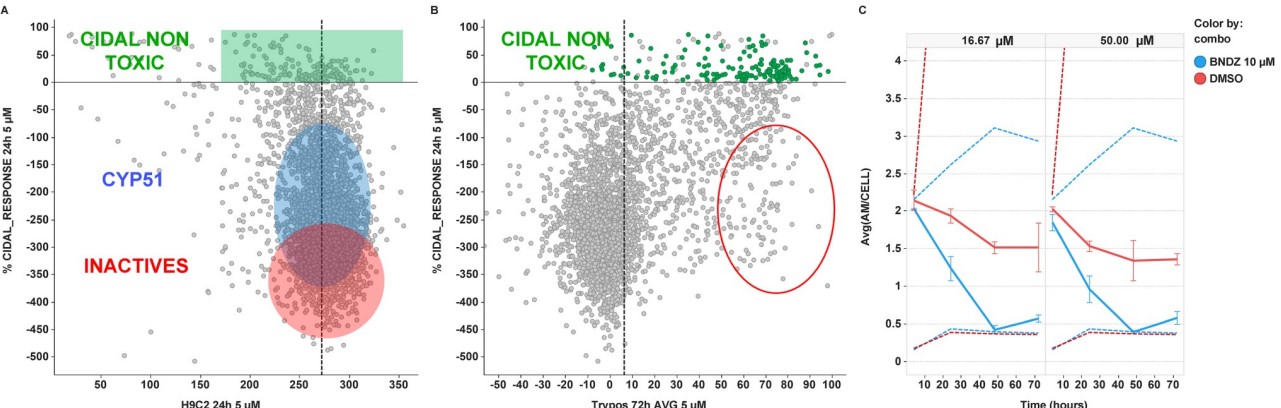

**Fig 6. Results of the 4,000 compounds screening campaign run against *T. crzui* RoK assay.** Compounds were tested at single-shot format (5 μM, 24 hours incubation). (A) Correlation plot between %CIDAL_RESPONSE against intracellular amastigotes and cytotoxicity, as number of H9c2 host cells *per* well. Three groups are highlighted: inactives in red, TcCYP51 inhibitors in blue and cidal *plus* non-toxic compounds in green. (B) Correlation plot between %CIDAL_RESPONSE against intracellular amastigotes and activity *versus* trypomastigotes; ellipse in red represents an arbitrary boundary to qualitatively illustrate compounds with prominent activity *versus* trypomastigotes but inactive against intracellular amastigotes (C) RoK time-course for combinations of Compound 1 (hit from the screening set) at 16.67 μM (left panel) or 50 μM (right panel) with either DMSO (red solid line) or BNZ (blue solid line) at 10 μM. Y axis represents average_AM/CELL output (AM/CELL: amastigotes *per* host cell). Dotted lines at the bottom of the plot correspond to non-infected cells controls, whereas dotted lines at the top of the plot corresponds to infected cells in the presence of either DMSO (in red) or 10 μM BNZ alone (in blue). Error bars represent the standard deviation of the average.

assay performed as described. Since BNZ at 50 μM reaches 100% maximum effect at 48 hours, only two plates at 4 and 24 hours were tested, whereas for non-cidal concentration of 10 μM, combinations were studied at 4, 24, 48 and 72 hours. As an example, RoK curves for one of the hit compounds selected are depicted in Fig 6C.

Compound 1 alone is cidal, but unable to clean all the parasites (red line). Remarkably, it can enhance the effect of the two selected concentrations of BNZ. Added with 50 μM of BNZ the effect of this concentration is faster, already cleaning up the parasites within the first 24 hours instead of 48 hours (S4 Fig). In the case of 10 μM of BNZ, the combination (solid line in blue) allows to reduce the concentration of BNZ by 5-fold to exert the same effect than 50 μM (dotted line in blue).

## Discussion

Despite the promising expectations raised from their *in vitro* and *in vivo* preclinical properties, clinical trials with ergosterol biosynthesis inhibitors, such as Posaconazole and the Ravuconazole prodrug E1224, rendered discouraging outcomes. Several different reasons have been argued as possible causes contributing to this failure [8, 11–14]: (i) suboptimal doses and/or duration of treatment due to an inappropriate PK/PD modelling or clinical design (ii) the usage of animal models of *T. cruzi* chronic infections which do not properly recapitulate Chagas disease in humans due to species-specific immunological interplay or differential life expectancy, and (iii) inability of azole compounds to eradicate all infective forms of *T. cruzi* in a reasonable time frame. In the current paper, we propose that fast-acting and cidal compounds have a better prospect for translation of *in vitro* activity into therapeutic efficacy. Hence, we have developed an experimental methodology for the triage of this kind of compounds as a way of increasing the probability of success in the selection of candidate molecules with activity in further efficacy models, providing they have the right PK/PD and ADME-TOX properties.

## Activity against trypomastigotes and intracellular amastigotes do not always correlate, and it is dependent upon the mode of action of the compound class

Whether an antichagasic candidate compound must exhibit activity against all possible live forms of *T. cruzi* to be efficacious in animal models and in the clinic is still arguable within the research community [17]. Nevertheless, here we explore the anti-trypomastigote activity as a surrogate assay to assess compound cidality. Trypomastigotes cultured *in vitro* are a non-dividing parasite form possessing a differentiated metabolic stage, thus only compounds targeting critical functions compromising cell viability will lead to parasite death. As previously described [17], POS was inactive *versus* trypomastigotes at both times whereas both BNZ and NIF were active with potencies in agreement with previously described pIC50 values (i.e. 5.3 and 5.9 for BNZ and NIF respectively), which remain invariable at the two incubation times tested (Fig 1A). Moreover, 61% of the compounds tested from the Chagas Box turned out to be inactive or hardly active against trypomastigotes. In other words, although all the compounds in the Chagas Box can abrogate or halt the proliferation of the amastigotes in the intracellular assay, only a small portion of those can eradicate the trypomastigote form with a reasonable potency.

Compounds with chemical groups such as imidazole, azoles, terminal free pyridines and aza-heterocyclic groups, which are already well described as inhibitors of cytochrome family through co-ordination to the haem iron (e.g. T. cruzi CYP51), were inactive in the assay [33, 34]. On the other hand, compounds containing nitroaromatic groups, e.g. TCMDC-143163, have a potent activity and a good correlation with the intracellular assay used for the TCAKS generation. The behaviour of these two chemical classes of antichagasic compounds are consistent with the results obtained with their corresponding standards and validate the trypomastigotes assay to distinguish CYP51 inhibitors in a large set of compounds.

## Discrete descriptors of kinetic patterns of Rate-of-Kill (RoK) assay can tell apart chemical and biological classes of antichagasic compounds

Early drug discovery programs tend to simplify the number and complexity of biological assays guiding identification and optimisation of chemotherapeutic compounds. Potency and efficacy values usually become the hallmarks of compound quality towards *in vivo* efficacy. However, this reductionist approach might be distorting the translational power of preclinical compounds. Hence, we propose to incorporate rate-of-kill (RoK) assays which allow us to characterise the speed and mode of action of compounds. Aiming at streamlining the analysis of kinetic patterns, we have defined simple discrete descriptors that can distinguish the nature of different chemical and biological spaces within large compound sets. As trypomastigote assay does, RoK curve descriptors are able to differentiate the two main pharmacological standards in Chagas Disease, i.e. azoles (POS) and nitroaromatics (BNZ and NIF) (Fig 2 and S2 Table). Furthermore, overall analysis of novel compounds out of the GSK Chagas Box, reveals a correlation between clusters of descriptors with chemical or biological class:

- **Group 1. Azoles and terminal pyridines**, which are known CYP51 inhibitors motifs, typically elicit characteristic bell-shape curves described by low ME values at 72 hours, negatives ME values at 24 hours, high t-lags, with no value in MCC50 but low MCCE values that are explained by their high potency as static drugs.

- **Group 2. Nitroaromatic compounds** used as reference (i.e. BNZ and NIF) show higher ME values close to 100% and short t-lag values. They are less potent than azoles, all the

active concentrations exhibiting cidal activity with not much difference between MCCE and MCC50.

- **Group 3**. Some characteristic chemical structures linked to **formerly reported mechanisms of action** are represented in the Chagas Box:

  - **Anti-folate** compounds containing a pteridine-like ring [35], such as TCMDC-143606 or TCMDC-143298, showed a kill curve characterised by high ME value and a tlag times of 4 hours. The potency of the compound in the standard imaging assay was high and it translates into low MCC and MCCE values.

  - TCMDC-140766 and TCMDC-139489 have analogue structures to previously reported **Methionyl-tRNA synthetase (MetRS) inhibitors** [36]. The descriptors of these compounds are similar in terms of cidality and speed of action with high ME at 24 hours and 72 hours, and short tlag times.
    Putative **MetRS and DHFR inhibitors** are able to kill trypomastigotes at 48 and 72 hours, similarly to nitroaromatic compounds, as aforementioned (Fig 1C and 1D).

  - Interestingly TCMDC-143137 show structural similarities with GNF5343 previously described **proteasome inhibitor** [37]. It only active in trypomastigote assay at 72 hours and in RoK assay exhibit a 100% ME at 72hours, but not signs of cidality at 24 hours. It shows similar tlag times than previously commented inhibitors.

  - Similar profile is shared by TCMDC-143083, **Formycin B, a inosine analog** with activity against *T. cruzi* [38]. It is cidal in the RoK assay, has ME positive for all times. It is inactive *versus* trypomastigotes at 48 hours but active at 72 hours.

  - The compound TCMDC-143087 is a known **inhibitor of the Qi active site of cytochrome b**, part of the cytochrome *bc1* complex of the electron transport chain (ETC) [39]. It is active *versus* trypomastigotes at 48 and 72 hours with the same potency and have a cidal profile with low values of tlag and positive value of ME from 24 hours. Nevertheless, it is not able to show cidal effect higher than 65% even at 72 hours.

- **Group 4**. Compounds like TCMDC-143331 or TCMDC-143610 are new structures with **unknown mode of action** that elicit kinetic pattern compatible with cidal mechanisms of action, i.e. high values in ME at the different times, short tlag and both are active in trypomastigotes assay at the two time points tested. Different profile is presented by the novel compound TCMDC-143623: cidal in the RoK assay, positive ME throughout all times and short tlag times, but poorly active in the trypomastigotes assay.

All in all, we propose that RoK and trypomastigote profiles correlate with chemical and biological series, suggesting that cidality and speed of kill are linked to mechanism of action. Compounds within the same chemical series are better differentiated by potency.

## Statistical clustering methodologies applied to RoK kinetic patterns provide a roadmap to guide compound optimisation programs

Looking at nine descriptors of the RoK curves to characterise the activity of a compound makes their comparison and classification cumbersome. With the aim of grouping compounds by their RoK patterns in a systematic and automated fashion, we have run unbiased clustering by three methodologies: principal component analysis (PCA) of the RoK descriptors (Fig 3, S1 and S2 Figs), k-means clustering of the full RoK experimental curves (Fig 4), and hierarchical clustering (Fig 5). As aforementioned in Results section, the three methodologies can

powerfully and appropriately discriminate azoles from nitroaromatics. The 2D-PCA plot divides the plane into four quadrants which are differentially populated by azoles and nitroaromatic compounds (Fig 3B). The outcome of the three methods for the pharmacological standards and the GSK Chagas Box compounds nicely correlates. Compounds belonging to k-means cluster 1 preferentially falls on top-right quadrant IV, e.g. CYP51/azoles (see also Fig 3C), whereas compounds in k-means cluster 2, e.g. nitroaromatic compounds, tend to populate bottom-left quadrant I.

Assuming that RoK pattern of nitroaromatic compounds have a higher probability of translating into *in vivo* efficacy, we propose that compounds falling on bottom-left quadrant I should have higher priority for selection than those ones falling on top-right quadrant IV. Therefore, we propose that a drug discovery program should pursue an optimisation trajectory towards compounds belonging to k-means cluster 2 falling on bottom-left quadrant I of the 2D PCA space. Remarkably, the compounds are distributed along a corridor from quadrant IV to quadrant I (Fig 3C). This space may well be used as the "optimisation trajectory" for a hit-to-lead or lead optimisation program in which new compounds are designed, synthesised, and tested *in vitro* prior to being selected for further *in vivo* studies of efficacy. Likewise, compounds falling on quadrant II of 2D-PCA are spread along the branches of the hierarchical clustering (Fig 5) on the way from red (azoles-like) to blue (nitroaromatic-like) clusters delineating a virtual "direction of travel", similarly to the one suggested from the PCA and k-means clustering (Figs 3C and 4B).

## RoK assays for the screening of large compounds set and combination studies

The experimental design of the RoK assay can be simplified to make it amenable to high-throughput format for applications aimed at triaging large sets of compounds along early drug discovery programs. Herewith we are showing its application for the screening of 4,000 compounds. For the sake of throughput, 5 μM single-shot concentration and one time-point of 24 hours have been selected as the most adequate conditions to discriminate static, such as CYP51 inhibitors, from cidal compounds. By correlating activity *versus* intracellular amastigotes with cytotoxicity, compounds can be grouped in at least three classes: (a) inactives, (b) putative static (azoles-like compounds), and (c) cidal but not cytotoxic (nitroaromatic-like compounds). Interestingly, there is a significant number of compounds exhibiting cidal activity *versus T. cruzi* intracellular amastigotes but no apparent effect on trypomastigotes assay even after 72 hours exposure. Contrarily, there exist compounds active against trypomastigotes while being inactive against intracellular amastigotes forms (Fig 6B, red ellipse). This observation agrees with the data represented in Fig 1D and with previously reported data by MacLean *et al.* (22). All in all, since (a) the lack of correlation between activities against trypomastigote and amastigote forms of the parasite, and (b) the latter being the most disease-relevant form of *T. cruzi*, we propose that trypomastigote assay is only used as a secondary assay to either flag CYP51 inhibitors or prioritise compounds with activity against both trypomastigotes and intracellular amastigotes forms of parasite.

Nitroaromatic compounds BNZ and NIF are currently the standard of care to treat Chagas Disease. Nevertheless, these drugs are handicapped by their adverse effects which may lead to the interruption of treatment in up to 20–30% of patients within a month [7]. Thus, combination regimens between BNZ and a second drug which reduces the dose and/or shortens the duration of treatment required to clean up the infection would be highly beneficial. In this regard, we have characterised the RoK profile of some pairwise combinations of BNZ with screening hits and compared with the compounds alone. Compound 1 was able to shorten the

time and reduce 5-fold the dose of BNZ to achieve maximum sterilising effect. Noteworthy, if this experiment had been carried out by using the standard imaging intracellular assay at 72 hours endpoint, as employed for potency determination, no enhancing effect of 50 μM BNZ combination would have been observed since maximum effect is reached at this concentration. Likewise, effects on slopes and asymptotes might be overlooked unless full time-course is monitored. Classical approach to combinations is based on simple IC50 calculations. However, RoK assays look at the mechanistic mode of action more holistically, thus increasing the chances to find enhancers. Killing kinetics approach has been formerly tested and proved valuable in *Plasmodium* for Malaria, *Mycobacterium* for tuberculosis, and *Leishmania* for visceral leishmaniasis [17, 40–42]. Seeking an eventual translation of an *in vitro* profile into *in vivo* efficacy, compound features impacting pharmacodynamics at the site of action cannot be neglected. Thus, we propose that *in vitro* RoK for *T. cruzi* is a cost- and time-effective tool to increase the probability of success in the selection of compounds which will translate *in vitro* activity into efficacy in animal models.

## Conclusions

Rate-of-kill (RoK) assays are amenable to high-throughput profiling, hence they can form part of a critical path for triage of large compound sets from screening hits, profiling of high-quality hits along the hit-to-lead or lead optimisation stages in drug discovery of anti-infectives. Herewith, we propose a simple, systematic and automated methodology of analysis of the otherwise complex kinetic patterns, i.e. PCA, k-means and hierarchical clustering, which provides drug discoverers with a roadmap to guide navigation along a compound optimisation program or prioritisation of best exemplars across different chemical series.

While sticking to the concept of interrogating the dynamics of the compound action against the parasite, the assay can be simplified and re-configured for the sake of increased productivity and throughput when large compound sets are profiled or screened, e.g. by running experiments at single concentrations and end-timepoint. Moreover, RoK studies shed light on the evaluation of drug combinations by potentially unveiling new enhancing features otherwise neglected, like speed of action or the maximum effect achieved.

As a result of an HTS campaign against the full 1.8M GSK collection, hundreds of hit compounds have been identified. We propose that speed of action for cidal effect, rather than potency, can differentiate those compounds with better prospect of success to show efficacy in animal models of Chagas disease. In the end, they might become right *in vitro* qualifiers to be brought in towards the prediction of an effective *in vivo* PK/PD profile.

## Supporting information

**S1 Table. Response in the trypomastigotes assay at 48 and 72 hours.** pIC50 = —log$_{10}$IC50 in molar units. Data from pIC50 values for CYP51 assay have been formerly reported [1]. (XLSX)

**S2 Table. RoK descriptors data for available compounds and standards.** Descriptors are computationally calculated as described in Methods. **ME** (Maximum Effect) at different time points (i.e. 24, 48 and 72 hours), **MCCE** (Minimal Concentration with Cidal Effect), **MCC50** (Minimal Cidal Concentration 50%), **t50%_MCC50** (time elapsed at MCC50 to reduce 50% of initial parasite load), **t50%_Cmax** (time elapsed at Cmax to reduce 50% of initial parasite load), **tlag_MCC50** (time elapsed at MCC50 to reduce parasite burden between two consecutive time points), **tlag_Cmax** (time elapsed at Cmax to reduce parasite burden between two consecutive time points) and **Cmin_Tox** (Minimal Cytotoxic Concentration). ND, not

determined. Threshold for colours are set as follows: **ME**, green for values higher than 50, yellow for values between 50 and -10 and red for values lower than -10; **MCCE**, green colour for values lower than 2 μM, yellow for values between 2 and 6 μM and red for values higher than 6 μM; **MCC50**, green for values lower than 5, yellow for values between 5 and 16 and red for values higher than -16; **t50%_MCC50**, **t50%_Cmax** and **tlag_Cmax**, green colour were used for values lower than 24 h, yellow for values between 24 and 48h and red for values higher than 48h; **tlag_MCC50** green for values lower than 24 h, yellow for 24h and red for values higher than 24h.
(XLSX)

**S1 Fig. Ordered eigenvalues (Scree plot) for PCA analysis of RoK curve descriptors.** Percentage of variance explanation for each dimension labels each bar. Dimension 1 and 2 covering 67% variance have been selected for further analysis.
(TIF)

**S2 Fig. Contribution of the RoK curve descriptors to 2D-PCA dimension 1 (A) and 2 (B).**
(TIF)

**S3 Fig. Schematic representation of the RoK curve descriptors.** (A-B) Data of Fig 2 were used to illustrate the RoK curve descriptors. In blue ME_72h, in red Tlag_Cmax, in yellow MCC50 and in purple T50pct_Cmax. Definition of all RoK curve descriptors are included in Methods section of the manuscript. (C) For MCCE schema non-normalised data (i.e. average of amastigotes per cell) were used Area in blue corresponds to the AUC of control sample in the absence of compound. The time-course line in green corresponds to the lowest compound concentration that reduces by 50% the AUC of control sample, i.e. MCCE. AUC for MCEE is coloured in light green.
(TIF)

**S4 Fig. Analysis of combinations of compound 1 with 50 μM BNZ.** Average of *T. cruzi* amastigotes per cell was plotted in Y axis and bars were coloured by concentration of compound 1 (50 μM in yellow, 16.67 μM in red and no compound in blue). 50 μM BNZ was present in all the cases. Two time-points were represented, i.e. 4 and 24 hours.
(TIF)

**S1 Code. Program coding in R language of the script for the computation of Rate-of-Kill curve descriptors.**
(R)

**S1 Raw Data. Raw data for figures and tables.**
(XLSX)

## Acknowledgments

Authors thank Vanessa Barroso, Imanol Peña and Carmen Punzon for their contributions to the development of experimental procedures and execution of experiments.

## Author Contributions

**Conceptualization:** Juan Cantizani, Ignacio Cotillo, Julio Martin.

**Data curation:** Juan Cantizani, Pablo Gamallo, Ignacio Cotillo, Raquel Alvarez-Velilla.

**Formal analysis:** Juan Cantizani, Pablo Gamallo, Ignacio Cotillo, Raquel Alvarez-Velilla.

**Investigation:** Juan Cantizani, Ignacio Cotillo, Raquel Alvarez-Velilla.

**Methodology:** Juan Cantizani, Pablo Gamallo, Ignacio Cotillo, Julio Martin.

**Project administration:** Julio Martin.

**Resources:** Juan Cantizani, Ignacio Cotillo, Raquel Alvarez-Velilla, Julio Martin.

**Software:** Pablo Gamallo.

**Supervision:** Julio Martin.

**Validation:** Juan Cantizani, Pablo Gamallo, Ignacio Cotillo.

**Visualization:** Juan Cantizani, Pablo Gamallo, Ignacio Cotillo.

**Writing – original draft:** Juan Cantizani, Pablo Gamallo, Ignacio Cotillo, Julio Martin.

**Writing – review & editing:** Juan Cantizani, Pablo Gamallo, Ignacio Cotillo, Raquel Alvarez-Velilla, Julio Martin.

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
