## [Decision Letter · Decision Letter 0]

21 Apr 2021

Dear Dr Martin,

Thank you very much for submitting your manuscript "Rate-of-Kill (RoK) assays to triage large compound sets for Chagas Disease drug discovery: Application to GSK Chagas Box" for consideration at PLOS Neglected Tropical Diseases. As with all papers reviewed by the journal, your manuscript was reviewed by members of the editorial board and by several independent reviewers. In light of the reviews (below this email), we would like to invite the resubmission of a significantly-revised version that takes into account the reviewers' comments. 

The manuscript was reviewed by experts in the field which agreed on its importance. However, the reviewers feel that there is space for much improvement regarding to concepts, as pointed by reviewer #3, and coherence, as pointed by reviewer #1, must be addressed by the authors. Improvement of methodological description and clarification of apparent results discrepancies need to be addressed by the authors. The submission of the program coding (if not protected by patent request) would be of great interest by the public.

We cannot make any decision about publication until we have seen the revised manuscript and your response to the reviewers' comments. Your revised manuscript is also likely to be sent to reviewers for further evaluation.

Sincerely,

Helton da Costa Santiago, M.D., Ph.D

Associate Editor

Helen Price

Deputy Editor

The manuscript was reviewed by experts in the field which agreed on its importance. However, the reviewers feel that there is space for much improvement regarding to concepts, as pointed by reviewer #3, and coherence, as pointed by reviewer #1, must be addressed by the authors. Improvement of methodological description and clarification of apparent results discrepancies need to be addressed by the authors. The submission of the program coding (if not protected by patent request) would be of great interest by the public.

Reviewer's Responses to Questions

**Key Review Criteria Required for Acceptance?**

**Methods**

-Are the objectives of the study clearly articulated with a clear testable hypothesis stated?

-Is the study design appropriate to address the stated objectives?

-Is the population clearly described and appropriate for the hypothesis being tested?

-Is the sample size sufficient to ensure adequate power to address the hypothesis being tested?

-Were correct statistical analysis used to support conclusions?

-Are there concerns about ethical or regulatory requirements being met?

Reviewer #1: See below

Reviewer #2: Most results are shown as averages and in some cases (Fig. 6C) error bars are shown but their meaning is not indicated. It would be important to have a section on Statistical analyses.

Reviewer #3: This manuscript describes the results of study aimed at proposing rate-of-kill (RoK) assays as a method to triage potential drug candidates for the etiological treatment of American Trypanosomiasis (Chagas disease, CD), a systemic parasitosis caused by the kinetoplastid parasite Trypanosoma cruzi. The aim is certainly of interest, as CD is the largest parasite burden in the American continent, now spreading to non-endemic areas due to international migrations and leading to severe cardiac and/or GI tract physiopathological alterations in ca. 30% patients. However, it is one of the most neglected diseases in the world as specific chemotherapy of this condition is very unsatisfactory, particularly in its prevalent chronic stage: the only drugs currently available in clinical settings, the nitroheterocyclic compounds benznidazole and nifurtimox, have limited and variable efficacy and frequent adverse effects, leading to treatment discontinuation in 10-30% of patients. As a result, it is currently estimated that <1% of patients receive treatment for this condition. 

Unfortunately, the motivation of the study is seriously questionable as its central premise is that fast activity (high RoK) is the key property to be prioritized in the development of anti-T. cruzi drugs (see page 6 and the Discussion), although it is well known that such compounds (including benznidazole and nifurtimox) are characterized by unspecific cytotoxicity that naturally leads to low selectivity and toxic side effects, in vitro and in vivo. Fast acting compounds typically act by what is called a Cmax effect, based on reaching a threshold concentration at the target organisms that triggers a rapid breakdown of essential structures and functions and cell death, while slow acting compounds act by specifically interfering with essential metabolic pathways in the target and require a sustained exposure, thus their activity is associated to the area under the curve (AUC; concentration vs time) of the drug, and are typically of low toxicity; for a recent experimental demonstration and discussion of this fundamental difference between the two types of drugs, see Meyer et al. (2019). J Antimicrob Chemother 2019; 74: 2303–2310.

**Results**

-Does the analysis presented match the analysis plan?

-Are the results clearly and completely presented?

-Are the figures (Tables, Images) of sufficient quality for clarity?

Reviewer #1: See below

Reviewer #2: Specific comments: 

Line 76: a reference is needed concerning nitroreductases

Line 89: a reference (Urbina, J. Eukaryot. Microbiol. 2015) suggested that those disappointing results were due to the use of posaconazole at a dose that was just 10-20% of that measured in mice at the curative dose of 20 mg/kg/d.

Line 477: “arguable within the research community”. A reference should be added or the sentence should be deleted.

Line 479: A better term than metabolically “poor” should be used or the term deleted. Although they do not divide, trypomastigotes are metabolically very active.

Reviewer #3: Yes.

**Conclusions**

-Are the conclusions supported by the data presented?

-Are the limitations of analysis clearly described?

-Do the authors discuss how these data can be helpful to advance our understanding of the topic under study?

-Is public health relevance addressed?

Reviewer #1: See below

Reviewer #2: Yes

Reviewer #3: In addition to the considerations presented above (Methods), the authors selectively cite the available scientific literature when claiming that ergosterol biosynthesis inhibitors, which act by selectively interfering with the synthesis of 24-alkyl sterols in fungi and protozoa such as T. cruzi and all species of Leishmania and thus are slow acting (AUC-depending) drugs, by claiming that lack of clinical efficacy of such compounds in recent clinical trials (page 5, refs. 4, 7-9) is associated to their mechanism of action, when the most plausible explanation for these disappointing results is the suboptimal doses and/or treatment duration used in these studies for the anti-T. cruzi indication (see Molina et al. 2015, Curr Opin Infect Dis 28:5, 397-407; Urbina, J.A. 2017. J Am Coll Cardiol. 11;70(2):299-300; Urbina, J.A. 2018. Lancet Infect Dis. 18(4):363-365). 

Thus, although the automated methodology described by the authors can nicely cluster separately fast acting from slow acting compounds, which correspond to Cmax (unspecific) and AUC (specific) agents (Fig. 3C and Results, pages 22-24), it is not a foregone conclusion that the former should be prioritized for further experimental and clinical development. On the contrary, such methodology could be very useful for an early characterization of potential candidates for anti-T. cruzi drugs with unknown mechanism of action as fast acting (Cmax) or slow acting (AUC) compounds, and thus raise red flags for potential in vivo toxicity and avoid long and expensive preclinical work with no clinical future.

**Editorial and Data Presentation Modifications?**

Reviewer #1: (No Response)

Reviewer #2: Minor revision

Reviewer #3: OK.

**Summary and General Comments**

Reviewer #1: In this manuscript, Cantizani et al develop quantitative metrics to identify cidal compounds against T. cruzi, in a format that is suitable for high-throughput analyses. This is a very important issue, with interesting new ways to visualize the data. Considerations of applicability to HTS are also welcome. However, I have the following major concerns:

1. MCC50 is defined by the authors as the lowest concentration that “reduces 50% of the parasite load measured at zero time.” However, POS is listed as having an MCC50 of 6.2 uM in Figure 2C, even though in Figure 2A there are no concentrations at which POS reduces parasite load by half. 

2. Likewise, I am puzzled as to how POS attains such a small MCCE given that it never reduces parasite load below ~1.5 and cidal response does not exceed 40%

3. MCCE for BNZ is calculated as 3.9 uM. I don’t understand how such a concentration could be considered to take 46.2 hours to clear 50% of parasites (t50%_MCC50), given that only the highest benznidazole concentration reduces amastigotes/cell compared to the starting load of ~2 ama/cell and other concentrations flatline or are associated with parasite growth in panel 2A.

4. t50%_Cmax is defined as the time needed for the highest concentration (50 uM) to reduce parasite load by half. Such a reduction (to ~1 ama/cell) seems to be attained before 20 h for nifurtimox, even though 24h is listed in Figure 2C. Likewise, I don’t understand how 65.9 is returned for this value for posa, given that the highest posa concentration doesn’t appear to attain a 50% reduction in parasite load even by experiment endpoint.

5. Benznidazole at the highest concentration always shows reduction in amastigote numbers at each timepoint. Why then is the tlag so high? I would expect it to be 0 at Cmax.

6. Nifurtimox is shown to clear parasites much faster than benznidazole based on Figure 2 curves; however, the difference in tlag between benznidazole and nifurtimox is quite small. tlag metrics should be improved or modified to reflect this difference.

7. These issues might be addressed by providing a schematic showing exactly how each value is derived from the am/cell and % cidal response curves.

8. Authors laudably talk about the usefulness of this method for HTS. The authors should provide their R script as supplemental data or as a github page. Otherwise, this method will not have any utility beyond the authors’ laboratory.

9. Lines 391-392, authors state “On one hand, all compounds belonging to the CYP51/azole class are scattered throughout clusters 1, 3, 4 and 5.” Close examination of cluster 2 also reveals some pale blue lines. Please clarify.

10. Figure 6B: “ellipse in red comprises compounds with activity versus trypomastigotes but inactive against intracellular amastigotes.” Why is this an ellipse and not a rectangular box? How were the margins selected?

11. Lines 470-471: “In the case of 10 μM of BNZ, the combination (solid line in blue) allows to reduce the concentration of BNZ by 5-fold to exert the same effect than 50 μM (dotted line in blue).”. I am puzzled by this statement. In figure 6C, the combination is showing much superior reduction in ama/cell compared to benznidazole alone, rather than showing the same effect. Please clarify.

Minor issues:

1. Line 132 typo: “sodiumpyruvate” should be “sodium pyruvate”

2. Figure 3C and 4A legends indicates “Color by TARGET”. However, the detailed legend is instead a mixture of compound name/class (e.g. NIF, nitroaromatic) and target (e.g. proteasome). Please make consistent.

3. Line 470 data not shown: please provide in supplemental

Reviewer #2: This is an interesting work proposing that speed of action and cidality, rather than potency, are better predictors of efficacy of compounds in animal studies, as suggested by previous results with azoles and nitroaromatic compounds. The method described could also be useful to triage large compound sets. This is a thoughtfully considered and well executed study. Overall, this manuscript advances the field significantly and I have only minor suggestions.

Reviewer #3: Based on the previous considerations, it is the opinion of this reviewer that the work is a very valuable effort in advancing the quest for safer and more efficacious chemotherapeutic anti-T. cruzi agents but the premise that originally motivated the study and the discussion of the experimental results presented in the manuscript must be profoundly revised, both on theoretical grounds and for eventual clinical applications of the methodology.

PLOS authors have the option to publish the peer review history of their article (what does this mean?). If published, this will include your full peer review and any attached files.

Reviewer #1: No

Reviewer #2: No

Reviewer #3: Yes: Julio A. Urbina
---

## [Decision Letter · Decision Letter 1]

28 Jun 2021

Dear Dr Martin,

We are pleased to inform you that your manuscript 'Rate-of-Kill (RoK) assays to triage large compound sets for Chagas Disease drug discovery: Application to GSK Chagas Box' has been provisionally accepted for publication in PLOS Neglected Tropical Diseases.

Best regards,

Helton da Costa Santiago, M.D., Ph.D

Associate Editor

Helen Price

Deputy Editor

Reviewer #1 found a possible inconsistency involving T50pct_Cmax, which we recommend the authors to double check during the production process.

Reviewer's Responses to Questions

**Key Review Criteria Required for Acceptance?**

**Methods**

-Are the objectives of the study clearly articulated with a clear testable hypothesis stated?

-Is the study design appropriate to address the stated objectives?

-Is the population clearly described and appropriate for the hypothesis being tested?

-Is the sample size sufficient to ensure adequate power to address the hypothesis being tested?

-Were correct statistical analysis used to support conclusions?

-Are there concerns about ethical or regulatory requirements being met?

Reviewer #1: (No Response)

Reviewer #2: Yes

Reviewer #3: Yes

**Results**

-Does the analysis presented match the analysis plan?

-Are the results clearly and completely presented?

-Are the figures (Tables, Images) of sufficient quality for clarity?

Reviewer #1: (No Response)

Reviewer #2: Yes

Reviewer #3: Yes

**Conclusions**

-Are the conclusions supported by the data presented?

-Are the limitations of analysis clearly described?

-Do the authors discuss how these data can be helpful to advance our understanding of the topic under study?

-Is public health relevance addressed?

Reviewer #1: (No Response)

Reviewer #2: Yes

Reviewer #3: Yes, in the revised version of the manuscript.

**Editorial and Data Presentation Modifications?**

Reviewer #1: (No Response)

Reviewer #2: Accept

Reviewer #3: None

**Summary and General Comments**

Reviewer #1: I appreciate the authors' efforts in clarifying the manuscript, and especially the additional figure (Figure S5) explaining the different metrics, which is very helpful. Overall, the manuscript is considerably improved and will be a valuable addition to the field. My last remaining concern is with regards to T50pct_Cmax in this figure (S5). The purple lines coming down to show the time readout on the x-axis are not coming down from the data curves in benznidazole in panel A and nifurtimox in panel B. Indeed, they appear to have been copy-pasted to exactly the same position for the benznidazole and nifurtimox curves, irrespective of the underlying data. For the posaconazole panels, the purple T50pct_Cmax lines appear to be intersecting with the curve error bars, rather than the curve itself. Please clarify.

Reviewer #2: Accept

Reviewer #3: In the revised version of the manuscript the authors have properly addressed the issues raised by this reviewer, as well as answering the questions and comments of the two other reviewers. As a consequence, it is my opinion that this revised version deserves publication in PLOS NTD.

PLOS authors have the option to publish the peer review history of their article (what does this mean?). If published, this will include your full peer review and any attached files.

Reviewer #1: No

Reviewer #2: **Yes: **Roberto Docampo

Reviewer #3: **Yes: **Julio A. Urbina

---

## [Editor Report · Acceptance letter]

10 Jul 2021

Dear Dr Martin,

We are delighted to inform you that your manuscript, "Rate-of-Kill (RoK) assays to triage large compound sets for Chagas Disease drug discovery: Application to GSK Chagas Box," has been formally accepted for publication in PLOS Neglected Tropical Diseases.

Best regards,

Shaden Kamhawi

co-Editor-in-Chief

Paul Brindley

co-Editor-in-Chief
